# LEARNING FROM IMPERFECTION: MISTAKE-AWARE LLM FINETUNING FOR ROBUST PLANNING

## ABSTRACT

While finetuned Large Language Models (LLMs) for embodied planning excel at producing reliable plans for the given environment, they do so in a very narrow area of operation. Usually one wrong step in such a plan is sufficient to get the agent into an unseen scenario. Current training paradigms focus on preventing mistakes by learning from optimal demonstrations, but neglect the crucial skill of recovering when deviating from the correct plan. To address this gap, we propose Mistake-Aware Finetuning (MAF), a novel training methodology that explicitly teaches agents to recover from planning errors. In the MAF paradigm, the model is exposed to plans containing intentional mistakes, but a targeted loss mask ensures it only learns from the subsequent, correct recovery actions. This allows the model to learn the association between a failure state and its resolution without being negatively influenced by the erroneous action itself. We demonstrate the effectiveness of MAF by finetuning a Llama-3B model on two distinct environments. Our approach substantially improves the task success rate from 21% to 72% on the Tower of Hanoi puzzle and from 67% to 96% on the complex Mini-Grid MiniBossLevel. Furthermore, to probe the generalisation capabilities of our method, we introduce Unlock-To-Unlock-N, a novel and challenging benchmark designed to test long-horizon planning. We demonstrate that our MAF-trained agent not only performs robustly in this new benchmark but also exhibits generalisation capabilities that extend beyond the training dataset. It highlights the MAF's immense potential for developing lightweight, robust embodied agents.

## 1 INTRODUCTION

Large Language Models (LLMs) have emerged as powerful planners for embodied agents, demonstrating a remarkable ability to translate high-level natural language instructions into executable action sequences (Wu et al., 2023; Huang et al., 2022; Song et al., 2023). However, their practical deployment is often hindered by a critical flaw: `brittleness`. In many scenarios, a single erroneous step can derail an entire plan, leading to unrecoverable failure (Valmeekam et al., 2023; Chakraborty et al., 2025). While sophisticated reasoning techniques like Chain-of-Thought can improve plan correctness, they often come with substantial computational overhead, rendering them unsuitable for many real-time applications (Zaremba et al., 2025).

The dominant paradigm in training such agents has focused on preventing errors by finetuning them on optimal, error-free plans. This approach is intuitive: to teach an agent to succeed, show it only success. While important, we argue this focus on prevention overlooks a crucial aspect of real-world intelligence: the ability to *recover from mistakes*. Even in predictable environments, LLMs having a hard time staying on the optimal path and therefore often deviate from it Huang et al. (2024). ***An agent that can only follow a perfect plan is an agent that will fail as soon as perfection is broken.***

This paper challenges the notion that learning must be confined to flawless demonstrations. We propose **Mistake-Aware Finetuning (MAF)**, a simple yet powerful training paradigm built on the central hypothesis that an agent becomes more robust by learning *from* mistakes, not just by avoiding them. In MAF, we intentionally include plans with errors in the model's input context but crucially exclude the mistake-generating steps from the training loss. The learning signal is focused solely on the subsequent *corrective actions*. This allows the model to learn the association between an error state and its correct resolution without being "poisoned" by the bad decision itself.

Figure 1: An overview of the Mistake-Aware Finetuning (MAF) data generation pipeline. An optimal plan (**Perfect Execution Steps**) is generated for a given environment and mission. A deliberate **Mistake Step** is then injected into this plan. This mistake leads to an error, at which point an optimal **Recovery Plan** is created. MAF trains the model to predict the recovery actions, using the mistake and its context as input but not as a training target.

When paired with a **step-by-step decoding** policy that generates one action at a time, MAF equips agents with the ability to dynamically replan and recover from unforeseen planning mistakes at execution time. We validate our approach across two distinct domains: the symbolic Tower of Hanoi puzzle and the interactive MiniGrid environment. Our contributions are:

- A robust finetuning method, **Mistake-Aware Finetuning (MAF)**, for teaching LLM-based planners to recover from errors. The core idea is to include intentionally flawed plans in the training data, where the mistake and its resulting error state serve as a rich learning context. Through targeted loss masking, the model is trained exclusively to predict the correct recovery sequence that follows, effectively learning from the *context* of a mistake without learning to *replicate* the mistake itself.

- The introduction of a new benchmark, **'Unlock-To-Unlock-N'**, created to test generalisation on complex, long-horizon tasks. The environment forces an agent to solve a recursive dependency chain, where it must first access N additional rooms in sequence to retrieve the keys needed to ultimately unlock the first door and complete its goal.

Shifting the training objective from simple error avoidance to effective error recovery offers a lightweight and scalable path toward more resilient LLM-based planners as shown by the significant performance gains, with success rates on Tower of Hanoi improving from 21% to 72% and on the MiniGrid MiniBossLevel from 65% to 94% (up to 96% with additional action information).

## 2 RELATED WORK

Our work builds upon several lines of research in LLM-based planning, error correction, and agent robustness. We position our contribution with respect to three key areas.

**LLMs as Grounded Planners.** Early explorations of Large Language Models (LLMs) for planning leveraged their zero-shot capabilities to generate action sequences from natural language goals (Huang et al., 2022; Valmeekam et al., 2022). A principal challenge, however, is their tendency to hallucinate infeasible actions that violate environmental constraints or preconditions. To address this, a significant body of work has focused on *grounding* LLM outputs. Methods like SayCan (Ahn et al., 2022) and SayCanPay (Hazra et al., 2024) filter the LLM's potential actions through learned affordance and value functions, ensuring that chosen actions are both possible and useful. Similarly, Grounded Decoding (Huang et al., 2023) integrates task constraints directly into the decoding process. While these techniques improve the validity of individual steps, they primarily focus on error prevention and do not equip the agent with a mechanism to recover once an error inevitably occurs.

**Static vs. Dynamic Planning.** Efforts to improve plan robustness have broadly followed two paths. The first involves *hybrid frameworks* that combine LLMs with classical symbolic planners. Approaches like ProgPrompt (Singh et al., 2022) and LLM+P (Liu et al., 2023) translate natural language goals into formal representations like PDDL, offloading the planning problem to a solver that guarantees correctness. The primary drawback of these methods is their rigidity; they rely on handcrafted domain models and produce static plans that offer little adaptability to unforeseen execution errors.

The second path is *dynamic replanning*, where agents generate plans incrementally (Suglia et al., 2021; Sharma et al., 2022). This step-by-step approach improves resilience to minor environmental shifts by re-evaluating the state at each decision point. However, these systems remain fundamentally brittle. An unexpected failure—such as trying to open a locked door without a key—typically halts execution, as the agent has not been trained to recognise or recover from such failure states. Our work embraces the flexibility of dynamic replanning but directly addresses this brittleness.

**Learning from Mistakes.** The most adjacent line of work explores whether models can learn from their own errors. LEMA (An et al., 2024), for instance, finetunes LLMs on datasets of errors and their corresponding human- or AI-generated corrections, demonstrating improved performance on reasoning benchmarks. While conceptually similar, our approach differs in two crucial ways. First, LEMA relies on external, often costly, supervision (e.g., annotations from GPT-4) to generate the "correct" behavior. Second, it is primarily evaluated on symbolic reasoning tasks and not in embodied environments. In this regard, some approaches have explored learning from error-prone trajectories with different data generation and learning strategies. ETO Song et al. (2024) collects contrastive pairs of successful versus failed trajectories through online exploration, applying DPO-style preference learning to distinguish them. IPR Xiong et al. (2024) uses Monte Carlo Tree Search to estimate step-level action values and creates contrastive pairs with outcome-level supervision from exploratory rollouts. AgentRefineFu et al. (2025) takes an LLM-based data-driven approach, using GPT-4o to generate synthetic error-prone trajectories followed by rule-based verification.

While these methods effectively leverage exploration or large-scale generation to discover diverse failure modes, they require substantial computational resources for online rollouts, MCTS search, or large model inference. In contrast, MAF operates offline with controlled mistake injection into expert trajectories, using targeted loss masking to directly supervise recovery behaviors. This eliminates the need for expensive exploration, value estimation, or external LLM supervision, while ensuring provably optimal recovery sequences through algorithmic experts. MAF's approach is particularly efficient for distilling recovery knowledge from capable but computationally expensive planners into smaller, deployable models.

## 3 MISTAKE-AWARE FINETUNING

In this section, we present the details of our core contribution, **Mistake-Aware Finetuning (MAF)**. The methodology is designed to teach planning agents recovery behaviors by augmenting the training data with examples of failures and their corresponding recovery plans. Crucially, we maintain a balanced curriculum: half of the training data consists of the original, optimal plans to ensure the model still learns to generate perfect plans.

The other half is augmented with mistake-induced plans. For these MAF samples, we mask the loss for the mistake-generating steps, using them only as input context. The training signal is focused exclusively on the subsequent recovery actions required to get back on track and complete the mission. This forces the model to learn the association between a failure state and its correct resolution without being "poisoned" by the bad decision itself. Furthermore, we propose a novel benchmark "Unlock-to-Unlock-N" in the MiniGrid environment suite to study the boundaries of planning agents in extrapolating to more complex long-horizon planning tasks.

### 3.1 DATA GENERATION FOR LEARNING RECOVERY BEHAVIORS

The core of Mistake-Aware Finetuning is a data augmentation process designed to transform a dataset of optimal plans into one that also teaches recovery from failure. For any given task, we generate a MAF-augmented data sample in three steps, as illustrated in Figure 2.

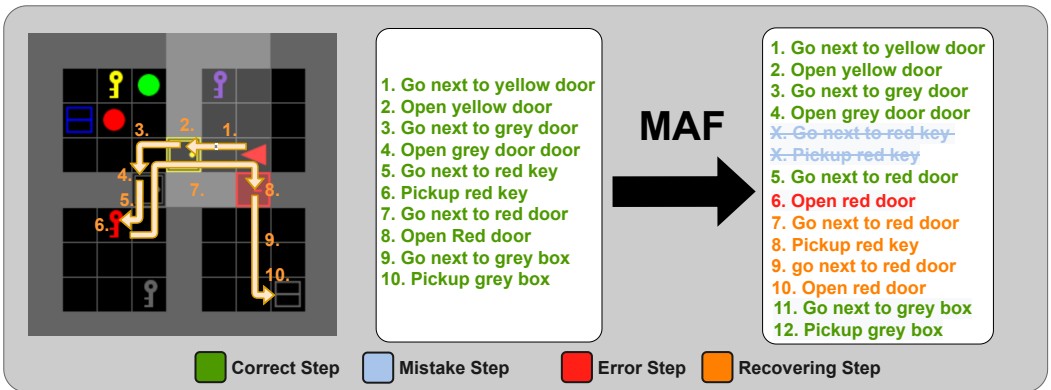

Figure 2: Visualisation of the Mistake-Aware Finetuning (MAF) data generation for MiniGrid's `MiniBossLevel`. Given an optimal subgoal trajectory. A mistake is then injected into this plan, leading to an error during execution (e.g., "The green door is locked"). From this error state, an optimal recovery sequence is generated. The MAF model learns to predict this recovery sequence, conditioning on the observed mistake and error state aswell as original perfect trajectories.

First, we use an expert planner to generate an optimal trajectory $\tau_{opt} = (s_0, a_0, s_1, \ldots, s_T)$, where $s_t$ represents the state and $a_t$ is the optimal action at step $t$. At a randomly selected step $k < T$, instead of executing the optimal action $a_k$, we inject a deliberate, suboptimal action $a'_k$. This action is chosen from a set of valid but incorrect moves, leading the environment to a new, non-optimal state $s'_{k+1}$. From this error state $s'_{k+1}$, we once again invoke the expert planner to generate a new optimal recovery plan, $\tau_{rec} = (s'_{k+1}, a_{k+1}, \ldots, s'_{T'})$, that completes the original mission.

The final training sample, $\tau_{aug}$, is a composite trajectory formed by concatenating the initial part of the optimal plan, the injected mistake, and the subsequent recovery plan:

$$\tau_{aug} = (s_0, a_0, \ldots, s_k, a'_k, s'_{k+1}, a_{k+1}, \ldots, s'_{T'})$$

This augmented trajectory contains a valuable learning signal. The sequence $(s_k, a'_k, s'_{k+1})$ serves as a demonstration of a mistake and its immediate consequence. The subsequent recovery plan $\tau_{rec}$ provides the crucial example of how to proceed correctly from that specific failure state. During finetuning, we leverage this structure through targeted loss masking to exclude the injected, suboptimal action from the training loss. This ensures the model is not "poisoned" by the bad decision and does not learn to replicate the error. The training signal is instead focused only on the subsequent recovery actions, effectively teaching the model to associate a given failure state with its recovery.

## 3.2 LEARNING PLANS WITH MISTAKE-AWARENESS

The data generated through our augmentation process forms a curriculum designed to teach resilience. The final training dataset is a balanced mixture of standard, error-free optimal trajectories ($\tau_{opt}$) and our MAF-augmented trajectories ($\tau_{aug}$). By training on both, the model learns not only to generate perfect plans but also to recognise and recover from common failure states.

The core mechanism of MAF is **targeted loss masking**. During the finetuning of the language model, which is a standard autoregressive, next-token prediction task, we selectively compute the training loss. For a standard optimal trajectory, the loss is calculated across all predicted actions. However, for an augmented trajectory $\tau_{aug}$, the loss associated with predicting the injected suboptimal action ($a'_k$) is masked and excluded from the gradient update. The learning signal is focused exclusively on the tokens corresponding to the corrective actions.

This ensures that the model learns the association between a failure context (the state $s'_{k+1}$ resulting from the mistake $a'_k$) and the appropriate recovery sequence ($\tau_{rec}$) without being "poisoned" by learning to replicate the error itself. By combining this training paradigm with a step-by-step decoding policy, the agent can dynamically replan and recover from unforeseen errors.

## 4 EXPERIMENTS AND RESULTS

Our experiments are designed to test the central hypothesis that training agents to recover from mistakes leads to more robust and generalizable planning capabilities. To this end, we directly compare our proposed Mistake-Aware Finetuning (MAF) approach against a standard baseline finetuned exclusively on optimal, error-free trajectories. We evaluate both methods across two distinct domains: the symbolic Tower of Hanoi puzzle, which tests logical consistency and long-horizon planning, and the embodied MiniGrid environments, which assess performance under partial observability and complex environmental constraints. Finally, we also explore the performance of MAF in comparison with preference-based offline RL for LLMs. This section outlines the specific implementation details of our models, describes the environments and data generation processes, and presents the results for both the baseline and MAF-trained agents.

### 4.1 IMPLEMENTATION DETAILS

All planning agents were developed by finetuning the Llama-3B model. The finetuning process was conducted for a single epoch for all experiments. For the embodied tasks, we utilised the MiniGrid environment to provide textual observations to the model. The expert data for MiniGrid was generated using the algorithmic `BabyAiBot` from SayCanPay Hazra et al. (2024)t.

The input to the language model was structured as a textual prompt containing the mission, a description of the current environment state, and the action history. The model was trained to output the next correct action or subgoal. For the step-by-step decoding strategies, the input was updated after each action, and in one variation, it was augmented with a list of all currently valid actions to ground the model's predictions.

### 4.2 ENVIRONMENTS

**Towers of Hanoi**   As an example planning task, we choose the classical problem of Towers of Hanoi puzzle consisting of multiple vertical rods and a corresponding number of disks of various diameters, which can slide onto any rod. The goal is to move disks one-by-one and stack them on top of each other in increasing order of their diameter, while ensuring that no disk is placed on another of smaller diameter at any step.

We use an optimal recursive algorithm to generate an optimal sequence of moves from any given state. At each step, we first identified the single best move. We then evaluated all other possible legal moves to determine their quality, specifically, whether they made no progress (redundant) or actively moved the game state further from the solution (recursively). We inject suboptimal yet valid moves into perfect plans with a **15% probability** at each step, wherein we force the agent to select one of these suboptimal moves. From this new, less optimal state, the optimal sequence is then computed. This process creates plans that contain deliberate, logical errors. During training, these mistake-aware plans are mixed in an even distribution with the original perfect plans. For the augmented data, the loss for the injected mistake step is masked out. The dataset, therefore, contains the suboptimal move and the resulting state in its input context, but the model is only trained to predict the correct, optimal recovery moves that follow.

**MiniGrid**   MiniGrid Chevalier-Boisvert et al. (2023) is a goal-oriented 2D discrete gridworld environment for evaluating planning capabilities of embodied agents on tasks of various complexities. Among the various environments, we choose the "MiniBoss" Level, which is a supersets of all other levels to test our approach at a high level of complexity. Environments can have multiple rooms to explore, which are seperated by doors, locked or unlocked. Additionally there are different objects the agent can pick up and drop. The observations are encoded into a so called "BabyLanguage" Chevalier-Boisvert et al. (2019) which can be seen in Section A.1

For the MiniGrid environment, we apply a similar principle of mistake injection but at the level of **high-level subgoal planning** (as shown in Algorithm 1). An expert agent, provided by Chevalier-Boisvert et al. (2019), first generates an optimal sequence of subgoals (e.g., `go next to green key`, `pickup green key`, `go next to green door`, `open green door`). To introduce a logical error, we programmatically corrupt this plan by removing a random subgoal of the plan or add a semantically fitting but random subgoal random into the plan. By adding a subgoal, the

plan must not necessarily be invalid, but has add a move which is not needed and therefore lengthens the plan. For instance, we would remove the `pickup green key` step, creating a flawed plan that instructs the agent to attempt opening a locked green door without acquiring the required key.

The agent then executes this corrupted plan until it inevitably fails and receives an error from the environment (e.g., "The green door is locked"). From this specific failure state, the expert planner is invoked again to generate a new, optimal recovery plan to complete the mission. Similar to the Tower of Hanoi setup, these mistake-and-recovery trajectories are mixed with pristine optimal plans. During finetuning, the model sees the flawed plan and the resulting error as input context but is only trained to predict the subsequent **corrective actions**, with the loss for the mistake-inducing steps being masked. This teaches the model to associate specific environmental failures with the appropriate recovery strategies.

**Unlock-to-Unlock-N**   To push the boundaries of our evaluation, we introduce a novel benchmark, designed specifically to test an agent's ability to generalise to longer-horizon problems. A common failure mode for planning agents is overfitting to the length and structure of training examples. This benchmark directly probes whether an agent has learned a generalizable, recursive strategy or has merely memorised fixed-length plans.

The setup is deceptively simple: the agent is instructed to pick up an object located in its starting room. However, this room is locked, and its key lies behind a chain of $N$ other locked rooms. Each room contains the key to the previous one, forcing the agent to solve a deep chain of dependencies: it must navigate to the Nth room, retrieve the key, unlock the (N-1)th door, and repeat this process until it can finally unlock the first door to retrieve the target object. As $N$ increases, the required plan length grows linearly, but the space of possible errors grows combinatorially, making it a severe test of long-term planning and recovery.

### 4.3 BASELINES

To establish a performance benchmark, we first train some baseline models using a standard finetuning approach on exclusively optimal, error-free data. These baselines serve as a direct comparison to quantify the performance gains from our Mistake-Aware Finetuning method in learning recovery behaviours.

**Tower of Hanoi.**   For the ToH environment, we generated a dataset of 10,000 optimal solutions for puzzles with 3 to 7 disks. A Llama-3B model was finetuned for a single epoch on this data. The training objective was to iteratively receive the current puzzle state, enumerate all legal moves, and then select the single correct move to advance the solution. When evaluated over 1,000 runs for each disk configuration, this baseline was notably brittle. It achieved a suc-

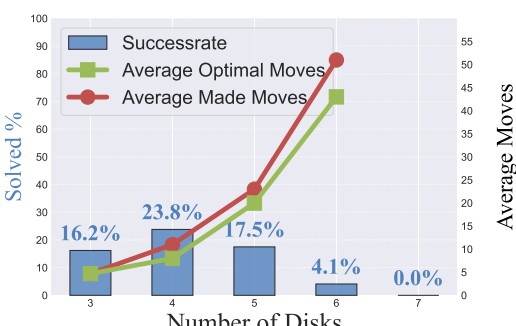

**ToH Precision and Efficiency: Baseline**

Figure 3: Evaluation of the Baseline ToH showing the Success Rate as well as the efficiency by average optimal moves and average made moves.

cess rate of only **21%** on puzzles with 3 to 5 disks. For more complex puzzles with 6 or 7 disks, the model consistently failed, as non-optimal moves caused the plan length to exceed generation limits.

**MiniGrid.**   For the embodied MiniGrid task, we generated 20,000 optimal plans using an expert agent on unique maps. A Llama-3B model was then finetuned to predict the correct sequence of high-level subgoals based on a given mission and a description of the level. We evaluated this baseline using three distinct decoding strategies on unseen maps: *(1) Full plan generation* yielded a 31% success rate, with failures often resulting from violated subgoal dependencies. *(2) Step-by-step decoding*, which provides immediate environmental feedback after each action, improved the success rate to 65%. *(3) Step-by-step with possible actions*, where the input at each step was augmented with a list of valid actions, grounded the model's decisions and established our strongest baseline with an 81% success rate.

## 4.4 RESULTS

In the following section we discuss the performance of our method on Tower of Hanoi, MiniGrid MiniBossLevel and Unlock-To-Unlock-N as compared to traditional finetuning techniques.

### 4.4.1 TOWER OF HANOI

To ensure a fair comparison, we finetuned and evaluated model using the identical experimental setup, parameters, and evaluation protocol as the baselines. The baseline model's performance was limited as shown in Figure 3; we report its 21% success rate only for disks 3-5, as attempts on 6 and 7 disks frequently exceeded token limits due to poor recovery strategies, such as getting stuck in circular paths or small loops. The MAF-finetuned model, however, did not suffer from this issue as regularly, demonstrating far greater robustness and achieving a 72% success rate across disks 3-5 as shown in Figure 4. While the baseline's successful runs have a lower average number of

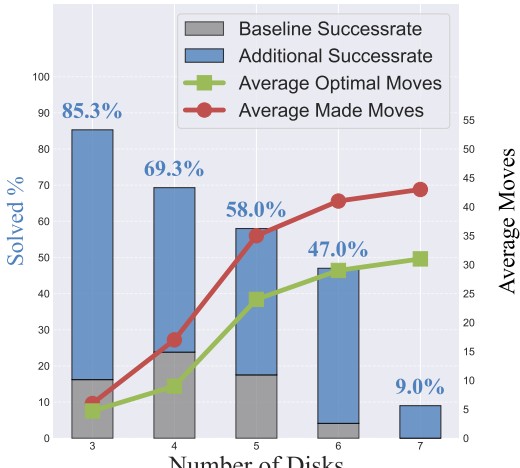

Figure 4: Evaluation of the Baseline ToH with MAF showing the Success Rate as well as the efficiency by average optimal moves and average made moves of successful evaluation runs.

moves, this is because it only managed to solve the simplest problems. MAF's higher move count reflects its ability to successfully navigate and recover from more complex, non-optimal states, which is a testament to its superior problem-solving capabilities.

| Decoding Strategy | Baseline | MAF |
|---|---|---|
| Full Plan | 31% | 36% |
| Step-by-Step | 67% | 79% |
| Step-by-Step with Possible Actions | 81% | **96%** |

Table 1: Success rates on MiniGrid's `MiniBossLevel`. We compare the performance of the Baseline training method against Mistake-Aware Finetuning (MAF) for all decoding strategies.

### 4.4.2 MINIGRID

**MiniBossLevel** In the embodied MiniBossLevel of the MiniGrid environment, mistakes were injected by corrupting the high-level subgoal plan, such as by removing a necessary prerequisite (e.g., removing 'pickup green key' before 'open green door'). The agent would execute this flawed plan until it failed, and the model was trained exclusively on the expert-generated recovery sequence. As shown in Table 1, MAF significantly improved performance here as well. For full-plan generation, the success rate saw a modest increase from 31% to 36%. However, the true impact was in the **step-by-step** setting, where the agent's ability to recover is transformative. The success rate surged from 65% in our baseline to **94%** with MAF. With the additional possible actions, the model achieved **96%**.

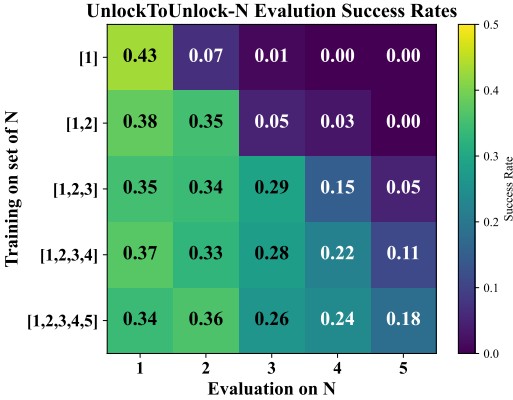

Figure 5: Generalisation on the `UnlockToUnlock-N` benchmark. We evaluate agents trained on task depths up to $N_{train}$ on unseen levels with depth $N_{test} \geq N_{train}$.

**Unlock-To-Unlock-N**  We took our best-performing agent, trained with step-by-step decoding, awareness, and MAF and finetuned it on a curriculum of 'UnlockToUnlock-N' tasks where $N \in \{1, ..., N\}$. We then evaluated its zero-shot performance on seen and unseen, more difficult instances where $N \in \{1, 2, 3, 4, 5\}$.

As shown in Figure 5, the results demonstrate an increasing capability of generalisation the higher the $N$, showing even in this hard recursive environment, generalisation is possible. Our MAF-trained agent maintained a decent success rate, being able to solve instances for unseen depths.

### 4.4.3 Ablation Study on the Proportion of MAF Data

To determine the optimal balance between learning from perfect plans and learning from mistakes, we conducted an ablation study on the **proportion** of MAF-augmented data in the training curriculum. We trained separate models on datasets with varying mixture ratios of MAF trajectories: 0% (the baseline), 25%, 50%, 75%, and 100%. Each model was then evaluated on the MiniGrid `MiniBossLevel` using the step-by-step decoding strategy.

The results, presented in Figure 6, show a distinct trend. Introducing MAF data provides a substantial performance boost, with the success rate climbing from **65%** at 0% MAF data to a peak of **94%** at a 50% mixture. This confirms that exposing the model to failure-and-recovery scenarios is highly effective for improving robustness.

However, the performance declines as the proportion of MAF data increases beyond this point, dropping to **89%** with a 75% mixture. A model trained exclusively on mistake-and-recovery trajectories (100% MAF) performed poorly, achieving only a **36%** success rate. This highlights a critical insight: while learning to recover is vital, it cannot replace the fundamental knowledge of the optimal policy. An agent

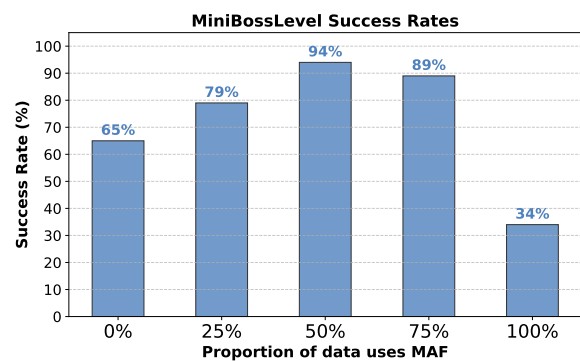

Figure 6: Effect of MAF data proportion on success rate in `MiniBossLevel`. Performance peaks at a 50% mixture, demonstrating the need for balancing both optimal and mistake-recovery examples.

trained only on fixing errors may not learn how to execute a plan correctly from a clean start. Therefore, a balanced curriculum, integrating both perfect demonstrations and mistake-aware recoveries, is essential for achieving maximum performance and robustness.

### 4.4.4 Performance with Preference-based Offline RL

To further benchmark our Mistake-Aware Finetuning (MAF) with established policy optimisation methods for LLMs, we implemented a two-stage training pipeline utilising Direct Preference Optimisation (DPO) Rafailov et al. (2023) on the `MiniBossLevel` of MiniGrid using a Step-by-Step approach. We compare simple supervised learning of mistake-aware recovery behaviours against an offline reinforcement learning objective that explicitly optimises for a preference between successful recovery and failure. The training process consists of two distinct sequential phases:

1. **Supervised Finetuning (SFT) Warmup:** We first train a base model using standard cross-entropy loss. This serves as the reference model for the subsequent RL stage. We experimented with two variants for this base model: one trained purely on optimal expert demonstrations, and one trained on our MAF-augmented dataset (SFT-MAF).

2. **Direct Preference Optimisation (DPO):** After the initial SFT phase, we continue training the policy using DPO Rafailov et al. (2023). In this stage, the model is trained on contrastive pairs of actions, namely, a "chosen" (preferred) action and a "rejected" (dispreferred) action. This objective forces the model to increase the log-likelihood of the preferred recovery trajectory while simultaneously decreasing the probability of the erroneous action, effectively unlearning the specific error pattern relative to the recovery.

**Preference Data Construction**   To adapt our planning task for the DPO format without an external reward model, we constructed preference pairs of chosen and rejected actions based on the intrinsic optimality of the plans. We utilise the erroneous actions from the MAF-augmented data as well as the subsequent steps until an error state is reached. All the actions along such trajectories from the deliberate suboptimal step leading to an error state are marked as a rejected sample, effectively assigning it a negative implicit reward. For every rejected sample, the corresponding chosen sample was defined as the expert recovery action that navigates from the error state back to mission success. Furthermore, the optimal actions recovering from the error state in the MAF-augmented dataset are paired against wrong, but viable, actions to reinforce maintaining optimality.

**Experimental Setups**   To evaluate the impact of our data augmentation strategy using mistake-aware recovery behaviours an offline RL context, we evaluated four specific training configurations of Supervised Finetuning followed by policy refinement using DPO:

- **SFT(Optimal) + DPO(Optimal):** We first compare a standard baseline where both stages focus exclusively on optimal demonstrations without any mistake recovery data.

- **SFT(MAF) + DPO(Optimal):** During SFT, we use our MAF data, but the DPO stage only reinforces preference for purely optimal paths against noisy actions.

- **SFT(Optimal) + DPO(MAF):** The model is initialised with data from only the optimal planner, with mistake-recovery data introduced solely during the DPO stage.

- **SFT(MAF) + DPO(MAF):** Both stages, SFT and DPO are trained on the MAF data containing mistake-recovery behaviours.

For a direct comparison and to control for total training compute, we also evaluated purely supervised baselines where the second stage consisted of continuing the SFT, which we denote as "SFT (Extended)". This ensures that any performance gains observed in the DPO variants are attributable to the preference optimisation objective rather than simply the extended training duration.

Table 2: Performance of Offline Preference-based RL using MAF Data.

| SFT \ DPO | Optimal | MAF | SFT (Extended) |
|---|---|---|---|
| **Optimal** | 82% | 85% | 78% |
| **MAF** | 90% | **92%** | 91% |

Our results indicate the effectiveness of introducing such mistake-aware recovery behaviours into the training data. This can especially be seen in the drastic performance difference between the model warm-started with just the optimal data (first row in Table 2) compared to using the MAF data (second row in Table 2) even when further refinement is done on just optimal data without the mistake-aware recovery behaviours. Moreover, refining a pre-trained model with DPO using the MAF data (second column in Table 2) shows better improvement that running DPO with only optimal data (first column in Table 2). An impressive result to note is that simply performing SFT on the MAF data with a larger compute budget performs similarly to refinement with DPO. This shows the importance of learning mistake-aware recovery behaviours.

## 5   DISCUSSION

Our experimental results provide compelling evidence in favor of our central hypothesis: teaching an agent to recover from mistakes is a crucial component for building robust planners. The standard paradigm of training solely on perfect demonstrations produces brittle agents that often fail upon the first deviation from the perfect path. Our work addresses this by embracing a seemingly paradoxical idea: by exposing the model to failure, we make it more successful. This section interprets why this approach is so effective, connecting our findings back to our core research question.

**Why Mistake-Aware Finetuning Works.**   The success of MAF is rooted in its ability to teach the model a critical skill that baseline agents never learn: associating a failure context with a corrective

action. A baseline model trained only on optimal paths has no reference for what to do when it enters a state resulting from an error (e.g., facing a locked door without a key). For such a model, this state is an out-of-distribution anomaly, leading to unpredictable and typically incorrect next actions. MAF, by contrast, explicitly provides these failure contexts as input during training. It teaches the model to recognise a suboptimal state in Tower of Hanoi or an unmet prerequisite in MiniGrid and directly maps this context to a high-probability recovery plan. It doesn't learn *to make* the mistake; it learns to recognise the mistake's aftermath and act accordingly.

**The Synergy of Recovery and Dynamic Replanning.** The results in the MiniGrid 'Mini-BossLevel' highlight a powerful synergy. While step-by-step decoding alone improves performance by giving the agent more opportunities to plan, it does not guarantee the agent knows *how* to use those opportunities after a failure. This is why our strongest baseline still plateaued at 81%. MAF provides the missing piece of the puzzle: the recovery knowledge. The combination of a dynamic setting (step-by-step) that provides the *opportunity* to recover and MAF that provides the *knowledge* for recovery proved transformative, pushing performance to 96%. This suggests that for embodied agents, robust performance emerges not from a single architectural choice, but from the interplay between the agent's decision-making structure and its training paradigm.

**From Memorisation to Generalisation.** An additional finding is the generalisation capability demonstrated on the 'Unlock-To-Unlock-N' benchmark. The MAF-trained agent, succeeded on this hard task even on unseen depths, because it learned a more abstract, recursive policy ("If a door is locked, find its key"). Because its training was punctuated with failures and recoveries, it was forced to learn the underlying causal logic of the environment rather than just mimicking a surface-level sequence of actions.

**MAF in RL.** Policy learning for agents interacting with environments is traditionally tackled in RL paradigms. However, applying online RL directly to Large Language Models presents significant hurdles: the immense sample complexity required for convergence is often prohibitive given the computational cost of updating billion-parameter models (Ouyang et al., 2022; Shinn et al., 2023). Consequently, while MAF serves as an efficient method for instilling robust priors, a promising avenue for future work lies in integrating value-based offline RL techniques. Approaches such as Implicit Language Q-Learning (ILQL) Snell et al. (2023), which enable reward-based learning from static datasets without the instability of online interaction, could effectively complement our pipeline. Combining such reward-driven offline RL mechanisms with preference-based finetuning represents an exciting direction for developing agents that are both resilient and sample-efficient.

**Limitations and Future Work.** Despite its success, our work has several limitations that open avenues for future research. First, our mistakes are generated synthetically. A key next step is to explore whether MAF is effective for recovering from errors that arise more organically, such as from noisy perception in the real world or from the model's own emergent failures during online learning. Second, we deliberately positioned MAF as a lightweight alternative to explicit reasoning methods like Chain-of-Thought. An exciting future direction would be to combine MAF with reasoning to investigate whether efficient recovery and deep reasoning can be complementary. Finally, we primarily focused on the agent's own planning errors. Extending MAF to handle errors arising from unpredictable environmental changes or multi-agent interactions remains a challenging but important frontier.

## REPRODUCIBILITY STATEMENT

To ensure the reproducibility of our results, we provide our source code and detailed experimental procedures. The supplementary material contains a zipped archive with the Python scripts used to generate the mistake-aware datasets for the MiniGrid environment MiniBossLevel. The provided code also includes the complete evaluation functions used to measure the success rates reported for both full-plan and step-by-step decoding strategies. The algorithmic process for generating the Tower of Hanoi data is described in Paragraph 4.2. All additional information like model specifications (Llama-3B), and finetuning hyperparameters are also detailed in the Appendix, providing a clear basis for replicating our findings.

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

## A  APPENDIX

---

**Algorithm 1:** MiniGrid MiniBoss Level Dataset Generation with Mistake-Aware Segmentation

---

**Function** `CreateData`(*envName, numSamples, save, savePath, datasetName, render, seedsPath, kwargs*)**:**

  $env \leftarrow$ **GymMake**(`BabyAI`$-envName, kwargs$);
  $env \leftarrow$ **LanguageObsWrapper**($env$);
  **for** $i \leftarrow 1$ **to** $numSamples$ **do**
    $obs \leftarrow$ `GetValidObservation`($env$);    // Filter invalid states
    $(entry, ok) \leftarrow$ `GenerateEpisode`($env$);    // Expert rollout with Recovery Agent
    **if** $ok$ **then**
      $segments \leftarrow$ `BuildSegments`($entry$); // Convert to MAF training format
      append $segments$ to dataset;
      **if** $save$ **then**
        write $segments$ to $savePath$

  **return** dataset

**Function** `GenerateEpisode`($env$)**:**

  $(obs, info) \leftarrow env.reset()$;
  $bot \leftarrow$ Recovery Agent($env$);
  $G \leftarrow$ GraphCreator();
  **while** *episode not finished* **do**
    $A_{\text{possible}} \leftarrow bot.get\_possible\_actions()$;
    $a \leftarrow bot.replan()$;
    $(obs, r, term, trunc) \leftarrow env.step(a)$;
    $G.add\_stack(bot.stack, A_{\text{possible}})$;
  $G.add\_end\_possible\_actions()$;
  **return** $\{\text{mission}, \text{level}, \text{plan}, \text{tree}, \text{possible actions}, \text{seed}\}$, $ok{=}true$;

---

### A.1  MINIGRID EXAMPLE EXECUTION

To further illustrate the hierarchical planning process, particularly for MiniGrid environments, we provide an example execution trace for a sample mission. The agent's goal is to open a specific door, which often requires navigating through a series of prerequisite subgoals.

**Mission and Level Description**  Mission: Open the blue door
Level: Room 1 has agent. Room 2 has yellow key, blue key. Room 3 has yellow box, green ball. Room 4 has grey ball, blue box, green box, yellow ball. The red door connecting Room 1 and Room 2 is closed. The grey door connecting Room 1 and Room 2 is closed. The blue door between Room 2 and Room 4 is locked.

The hierarchical decomposition of this mission, as constructed by the agent's goal stack, is visualised in Figure 8. The agent dynamically breaks down the high-level mission into smaller, manageable subgoals, each becoming a node in this tree. The plan unfolds by addressing the deepest (most dependent) subgoals first, moving upwards towards the ultimate mission completion.

**Execution Steps**  The agent successfully completes the mission by executing the following sequence of high-level subgoals:

1. Go next to grey door
2. Open grey door
3. Go next to blue key
4. Pickup blue key

5. Go next to blue door

6. Open blue door

This sequential completion of subgoals leads to: Mission Complete.

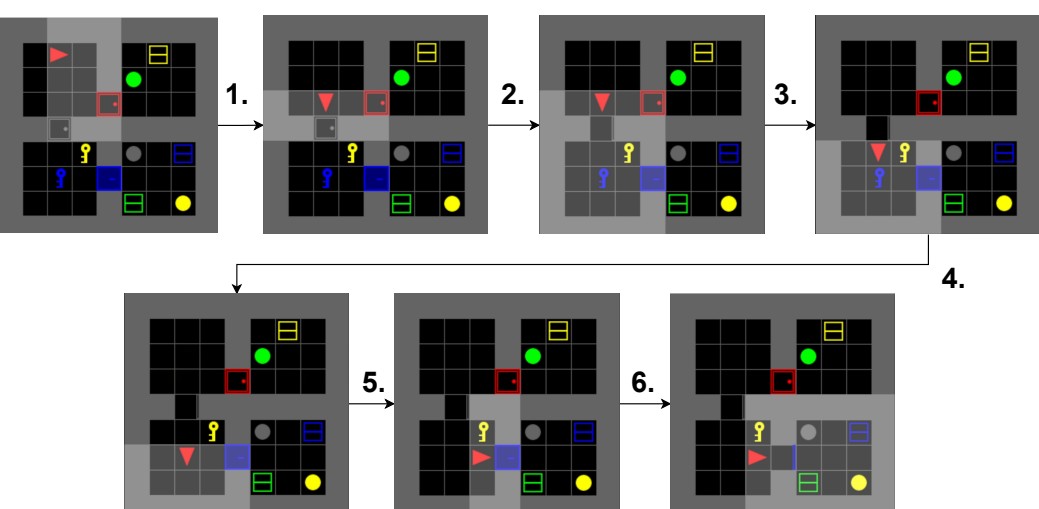

Figure 7: Step-by-Step Visualisation of a Minigrid Environment with the mission: "Open the blue door"

**Hierarchical visualisation**   While creating the optimal solution, we generate a hierarchical representation of the task, as shown in Figure 8. This tree is a direct visualisation of the agent's goal stack at its most complex point; each level of depth corresponds to a new subgoal pushed onto the stack to resolve a dependency. The deepest nodes, or leaves, represent the most immediate prerequisites that must be solved first. The chronological plan to complete the mission is therefore derived by accomplishing the subgoals from the leaves upwards, as each child task is a prerequisite for its parent.

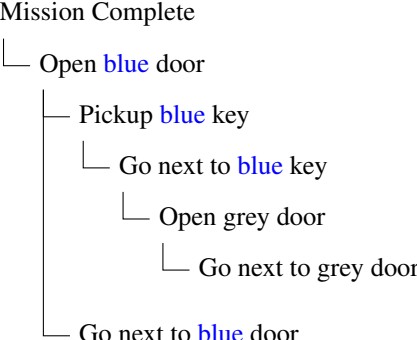

Figure 8: Hierarchical subgoal tree for the "Open the blue door" mission, illustrating the agent's goal stack decomposition.

**Hyperparameters**

A.2   TRAINING AND HYPERPARAMETER DETAILS

The finetuning process was conducted on a high-performance compute node equipped with 4x NVIDIA H100 GPUs, each with 80GB of VRAM.

The primary goal of the finetuning was to adapt the pre-trained Llama-3.2-3B model to the task of sequential action planning and recovery. The complete set of hyperparameters used for all experiments is detailed in Table 3. We employed a global batch size of 16, achieved through a per-device micro-batch size of 2 and 2 gradient accumulation steps across the 4 GPUs. A cosine learning rate scheduler with a brief warmup period was used to ensure stable convergence. For numerical stability and performance, we used bfloat16 mixed-precision training. The DeepSpeed ZeRO Stage 1 optimisation strategy, combined with the PyTorch 2.0 compiler, was used to manage memory and computational load effectively.

Table 3: Finetuning hyperparameters and training configuration.

| Hyperparameter | Value |
|---|---|
| Base Model | `meta-llama/Llama-3.2-3B` |
| Precision | `bf16` (bfloat16) |
| Max Sequence Length | 4096 tokens |
| *Training Parameters* | |
| Epochs | 1 |
| Learning Rate | $5 \times 10^{-5}$ |
| LR Scheduler | Cosine Decay |
| Warmup Steps | 100 |
| Weight Decay | 0.1 |
| Micro Batch Size (per GPU) | 2 |
| Gradient Accumulation Steps | 2 |
| **Effective Global Batch Size** | **16** ($2 \times 2 \times 4$ GPUs) |
| GPUs Used | 4x NVIDIA H100 (80GB VRAM) |
| DeepSpeed Strategy | ZeRO Stage 1 (`zero1_torch_compile`) |

