# OpenReview forum: "Learning from Imperfection: Mistake-Aware LLM Finetuning for Robust Planning"
_ICLR.cc/2026/Conference — Submitted to ICLR 2026_

### Official Review · Reviewer_LQcp · 2025-10-16

**Soundness:** 1
**Presentation:** 2
**Contribution:** 2
**Rating:** 2
**Confidence:** 4

**Summary:**

This paper focuses on learning from imperfect demonstrations where some mistakes is made at certain step of the trajectory. The article proposed a data generation method to construct a data combines with optimal and imperfect trajectories. Then a targeted loss masking is used to train the LLM to recover from errors. To validate their method, the author proposes a new benchmark: Unlock-to-Unlock-N and does some experiments.

**Strengths:**

1. The idea of learning from imperfect demonstrations is novel.
2. The author proposes a new benchmark: Unlock-to-Unlock-N.

**Weaknesses:**

1. The setting is impractical. The method requires to know at which step the mistake is made in advance, while in many practical settings it is impossible.
2. The method is too simple. Only ignore the mistaken step to train is just a simple weighted behavior cloning with weights only being 0 and 1. We desire a more innovative and effective approach. For example, some offline RL methods can weight these behaviors with a learnt Q.
3. The baseline is too simple and not enough. For example, you can include an offline RL baseline with the right step rewarded by 0, wrong step rewarded by -0.1 and final success rewarded by 1, where the policy is learnt with a Q as weighting function on the behaviors.

**Questions:**

See weaknesses.

---

> ### Author Response · Authors · 2025-11-28
>
> > The setting is impractical. The method requires to know at which step the mistake is made in advance, while in many practical settings it is impossible.
>
> We apologize that this aspect was unclear in our explanation. We believe there is a misunderstanding about when mistake identification occurs. During training, we know which steps are the mistake steps since we injected them manually. This can be thought of as a parallel to the exploration-exploitation tradeoff in RL, where random actions are sampled for exploring the state space. Since a known mistake step and subsequent recovery actions are added to the dataset during training, we assume that during inference, even if an erroneous step is taken, the behaviour generated by our model implicitly learns to recover from it. This can be seen in our ablation study (Figure 6), wherein adding such mistake-aware data to the dataset to implicitly learn recovery behaviours helps improve the overall performance during inference, **without any special error-detection mechanism**. We thank the reviewer for giving us the opportunity to clarify this part of our approach. We shall additionally improve the explanation in the manuscript, which we shall update soon.
>
> ---
>
> > The method is too simple. Only ignore the mistaken step to train is just a simple weighted behavior cloning with weights only being 0 and 1. We desire a more innovative and effective approach. For example, some offline RL methods can weight these behaviors with a learnt Q.
> >
> >
> > The baseline is too simple and not enough. For example, you can include an offline RL baseline with the right step rewarded by 0, wrong step rewarded by -0.1 and final success rewarded by 1, where the policy is learnt with a Q as weighting function on the behaviors.
>
> We thank the reviewer for this valuable suggestion. As also mentioned by the other reviewers, we are currently running experiments with Offline RL comparison, and shall provide an update soon.

---

### Official Review · Reviewer_MfaT · 2025-10-28

**Soundness:** 4
**Presentation:** 3
**Contribution:** 3
**Rating:** 6
**Confidence:** 5

**Summary:**

This paper presents a fine tuning technique for agentic planning tasks which makes the model more robust after a mistake.

**Strengths:**

The novelty in the paper is that the mistakes are introduced artificially during training and since they know which step is noisy, they do not consider that for model training (like masking). This way model learns how to robustly recover from past mistakes. The results are verifying the uselfulness of this approach on multiple benchmarks.

**Weaknesses:**

The title is misleading. It reads like this is an approach which is learning from its mistakes (like RL/DPO does), hence model makes less mistakes. The goal is not making less mistakes but being more robust after making a mistake. In agentic world, this is usually called as error recovery. Otherwise learning from mistakes is a heavily studied area with multiple papers (see below)
One big problem in the setup is that, the introduced mistakes do not effect the remaining actions. For example picking up an apple is introduced as a mistake but this can be ignored during the execution of the rest. In many cases the mistakes result in scenarios which are very hard to recover from, such as boiling an egg while making an omelette in home robot scenarios. Also for conversational cases, usually a mistake is harder to recover from, such as "set a timer" "for how long?" "15 minutes" "ok setting the timer for 50 minutes" "no i said 15 15"
In that respect I find the evaluation setup limited to certain tasks.

**Questions:**

Despite allocating a section on RL, I'd prefer to see quantifiable comparison with RL based approaches.
Explain the difference of your approach from the following papers:
https://arxiv.org/pdf/2403.02502
https://arxiv.org/abs/2406.11176
https://arxiv.org/pdf/2501.01702

---

> ### Author Response · Authors · 2025-12-03
>
> > Despite allocating a section on RL, I'd prefer to see quantifiable comparison with RL based approaches.
>
> We thank the reviewer for this valuable suggestion. As also mentioned by the other reviewers, we are currently running experiments comparing MAF with Offline RL. We shall provide an update shortly with the results.
>
> ---
>
> > Explain the difference of your approach from the following papers:
> >
> > 1. https://arxiv.org/pdf/2403.02502
> > 2. https://arxiv.org/abs/2406.11176
> > 3. https://arxiv.org/pdf/2501.01702
>
> We thank the reviewer for pointing out these relevant papers. Here are the key distinctions:
>
> **vs. ETO (Exploration-based Trajectory Optimization)** [[1]](https://arxiv.org/pdf/2403.02502)
>
> ETO learns from contrastive pairs of successful vs. failed complete trajectories collected through trial-and-error, using failed attempts as negative examples. MAF differs fundamentally in its data generation and learning mechanism: we programmatically inject mistakes into optimal plans and use targeted loss masking to exclude erroneous actions from the training signal while retaining them as context. Moreover, ETO works in an online manner by applying DPO-styled preference-based learning on failed rollouts from the current policy. MAF, on the other hand, operates in an offline manner by synthetically generating mistake-recovery pairs from a single expert planner, making data generation more efficient and controllable.
>
> **vs. IPR (Iterative Process Refinement)** [[2]](https://arxiv.org/abs/2406.11176)
>
> While both MAF and IPR use step-level refinement, IPR does so via Monte Carlo Tree Search to estimate step-level action values and creates contrastive pairs of better vs. worse actions. IPR additionally uses outcome-level supervision from the results of explorative trajectories as an extra supervisory signal. In contrast, MAF takes a more direct approach by programmatically injecting mistakes and using algorithmic experts to generate recovery sequences. Specifically, MAF doesn't require expensive MCTS exploration or value estimation during data generation. We directly create mistake-recovery pairs where the mistake context is preserved but masked from the loss, while the expert recovery sequence provides the learning signal. This makes MAF significantly more scalable.
>
> **vs. AgentRefine** [[3]](https://arxiv.org/pdf/2501.01702)
>
> While both MAF and AgentRefine adopt mistake-aware training paradigms to enhance agent generalization, they differ fundamentally in their data generation and subsequent generalization. MAF injects controlled, predefined mistakes and employs a deterministic expert planner to generate the optimal trajectories, ensuring the recovery paths are provably optimal. In contrast, AgentRefine embraces a synthetic, LLM-driven methodology where GPT-4o generates error-prone trajectories and reasonings, thereby relying on post-hoc rule-based verification of the generated trajectories. Essentially, MAF represents a more precise approach leveraging domain-specific expertise to teach recovery from known failure modes, while AgentRefine adopts a large-scale data-driven approach that uses emergent error-correction behaviors over algorithmic optimality.

---

### Official Review · Reviewer_gYwe · 2025-10-29

**Soundness:** 2
**Presentation:** 2
**Contribution:** 2
**Rating:** 2
**Confidence:** 3

**Summary:**

This paper tackles the brittleness of LLM-based planning agents, where a single error can derail the entire plan, by introducing Mistake-Aware Finetuning (MAF). Unlike conventional fine-tuning approaches that rely solely on perfect demonstrations, MAF intentionally injects errors into plans and masks the corresponding loss for those erroneous steps. The model is then trained only on the subsequent corrective actions, enabling it to associate failure states with appropriate recovery behaviors. Experiments are conducted on two domains: i) Tower of Hanoi, representing symbolic, logic-based planning, and ii) MiniGrid MiniBossLevel, representing embodied, partially observable environments. MAF significantly improves task success rates: from 21% to 72% on Tower of Hanoi and 65% to 94% (up to 96%) on MiniGrid. Additionally, the paper introduces a new benchmark, Unlock-To-Unlock-N, to evaluate long-horizon generalization, demonstrating that MAF-trained agents can generalize to unseen planning depths.

**Strengths:**

- Clarity and Motivation: The paper clearly defines the brittleness problem in LLM-based planners and convincingly argues for shifting the learning objective from error prevention to error recovery. This conceptual pivot is both intuitive and novel.
- Methodological Simplicity: MAF requires only minimal modification (loss masking) to existing fine-tuning pipelines. Its lightweight design supports scalability and ease of integration into existing LLM-planning frameworks.
- Empirical Rigor: Consistent performance gains are demonstrated across symbolic (Tower of Hanoi) and embodied (MiniGrid) tasks. The inclusion of the Unlock-To-Unlock-N benchmark provides evidence for the model’s generalization to long-horizon planning scenarios.
- Analytical Depth: The paper thoughtfully interprets MAF not merely as “error correction” but as learning a mapping between failure contexts and corrective actions. The synergy between MAF and step-by-step decoding is particularly well-analyzed, reinforcing the method’s robustness.

**Weaknesses:**

- Limited Experimental Settings: All experiments are conducted in symbolic or textual simulation environments (Tower of Hanoi and MiniGrid), which lack sensor noise, actuation errors, or physical constraints. As the authors themselves acknowledge in Section 5, MAF’s effectiveness in embodied or multimodal agents remains untested. The current validation is confined to text-level proxies of planning, rather than full embodied intelligence.

- Synthetic Mistakes: The mistakes used for training are artificially injected (e.g., random subgoal removal or insertion in MiniGrid, suboptimal moves in Tower of Hanoi). Such synthetic perturbations do not fully capture real-world error modalities (e.g., perception noise, actuator drift, memory errors, or environmental dynamics). Consequently, it is unclear whether MAF’s recovery capability would transfer to more realistic, noisy embodied scenarios.

- Lack of Comparison with Related Baselines: Although the paper conceptually relates MAF to lightweight alternatives to reinforcement learning (RL) approaches such as [1,2,3,4,5,6,7], it does not include quantitative comparisons or efficiency analyses against these baselines. Without such evaluations, it remains uncertain whether MAF’s masking-based fine-tuning provides a substantial benefit beyond existing (offline) RL or trajectory relabeling methods.

- Underdefined “Mistake-Aware” Taxonomy: The paper promotes “mistake awareness” as a central contribution but does not systematically define or categorize different error types. The injected mistakes are limited to planning-level symbolic errors, while real embodied agents may face more diverse and interacting failure sources (e.g., perceptual, causal, or motor-level mistakes). The lack of error taxonomy and per-type recovery analysis makes the notion of “mistake-awareness” somewhat shallow and limited to simple plan deviations.

[1] Off-policy deep reinforcement learning without exploration. Fujimoto et al., 2019.

[2] Conservative Q-Learning for Offline Reinforcement Learning. Kumar et al., 2020.

[3] AWAC: Accelerating Online Reinforcement Learning with Offline Datasets. Nair et al., 2020.

[4] Offline Reinforcement Learning with Implicit Q-Learning. Kostrikov et al., 2021.

[5] Decision Transformer: Reinforcement Learning via Sequence Modeling. Chen et al., 2021.

[6] Offline Reinforcement Learning as One Big Sequence Modeling Problem. Janner et al., 2021

[7] Hindsight Experience Replay. Andrychowicz., 2017.

**Questions:**

- How would MAF perform in multimodal or embodied settings involving perception and motor control noise? Do the authors consider sensor-based observations or continuous control domains in the current submission?

- Can the authors provide a more systematic definition or taxonomy of “mistakes” beyond subgoal insertion/removal? How might MAF handle errors such as perceptual, causal, or memory origin?

- Have the authors considered benchmarking MAF against offline RL methods, which similarly learn from suboptimal trajectories? How does MAF compare in terms of sample efficiency or robustness?

- Does this submission consider integrating MAF into larger embodied systems (e.g., robot task planners using visual-linguistic inputs)? If so, how are non-symbolic (or textual) state representations addressed?

---

> ### Author Response · Authors · 2025-11-28
>
> > How would MAF perform in multimodal or embodied settings involving perception and motor control noise? Do the authors consider sensor-based observations or continuous control domains in the current submission?
>
> Thank you for this important question. While our current work focuses on symbolic planning environments, we acknowledge that real-world embodied agents face additional challenges from perceptual and motor noise. We view MAF as complementary to perception modules rather than a complete solution. In our framework:
>
> - **Perceptual errors** (e.g., misidentifying objects) would manifest as unexpected observation feedback, which MAF-trained agents can learn to handle through recovery
> - **Motor noise** (e.g., imprecise movements) similarly produces error states that trigger the recovery mechanism
>
> For continuous control domains, the core MAF principle, i.e. learning from failure contexts, remains applicable, though the implementation would require modifications to handle continuous action spaces. Future work could integrate MAF with techniques like behavioral cloning from observations, or, for example, in a similar manner as [RoboMonkey](https://robomonkey-vla.github.io/) by augmenting trajectories with noise to subsequently pursue preference-based learning.
>
> ---
>
> > Can the authors provide a more systematic definition or taxonomy of “mistakes” beyond subgoal insertion/removal? How might MAF handle errors such as perceptual, causal, or memory origin?
>
> Our current work focuses on **action-level mistakes** (selecting wrong/suboptimal actions), but we acknowledge that a broader taxonomy should exist when considering real-world embodied agents. The key insight is that MAF's loss-masking approach generalizes: as long as we can identify error tokens and have expert recovery demonstrations, we can apply the methodology to distill the expert behaviour into a student model.
>
> - We do not explicitly consider memory errors, since we do not introduce any previous state information. However, we do keep the previously executed action history as part of the inputs. While this could cause issues with larger context lengths in long-horizon tasks, the performance of MAF on the proposed “UnlockToUnlock-N” benchmark (Figure 5) shows that even for larger environments with multiple rooms, our approach maintains a decent success rate.
> - Causal errors, such as incorrect beliefs about action effects, are partially addressed since such errors that can occur and subsequent recovery behaviours are part of the process of data generation for MAF to learn correct state transitions.
> - Perceptual errors, such as false detections in Task and/or Motion Planning scenarios for embodied agents, can be addressed through further data augmentation. This involves injecting similar mistakes into the training data to incorporate recovery behaviors caused by such errors. To some extent, such errors are covered by our framework, for example, by injecting wrongly coloured objects into the mistake actions, which in real-world settings could result from errors in perception and state estimation.
>
> ---
>
> > Have the authors considered benchmarking MAF against offline RL methods, which similarly learn from suboptimal trajectories? How does MAF compare in terms of sample efficiency or robustness?
>
> We thank the reviewer for this valuable suggestion. Offline RL methods do learn from suboptimal trajectories, however typically require a larger amount of data to train a critic well to differentiate between suboptimal and optimal actions. We are currently performing experiments comparing MAF with Offline RL and will update our submission and the comments with the results shortly.
>
> ---
>
> > Does this submission consider integrating MAF into larger embodied systems (e.g., robot task planners using visual-linguistic inputs)? If so, how are non-symbolic (or textual) state representations addressed?
>
> MAF operates at the action-planning level and is agnostic to state representation. This modularity is actually demonstrated in our MiniGrid experiments, where the environment provides "BabyLanguage" observations (structured text). For truly end-to-end visual learning, MAF can be combined with vision-language models (VLMs) and semantic scene graph representations, which have been shown to perform well for task planning scenarios on embodied agents in the real world, such as with [SayPlan](https://sayplan.github.io/), [DELTA](https://delta-llm.github.io/), [AEG](https://davit666.github.io/AEG-rearrangement/).

---

### Author Response · Authors · 2025-12-03
**Updating the paper with results from offline preference-based RL**

As requested by the reviewers, we have explored the performance of offline RL. Specifically, we investigate the use of [Direct Preference Optimization (DPO)](https://arxiv.org/abs/2305.18290), which is a preference-based RL approach with an implicit reward understanding. We have updated the results in the new version of our paper accordingly (Sec. 4.4.4).Specifically, we implemented a two-stage training pipeline using Direct Preference Optimization (DPO), where preference pairs are constructed from our mistake-augmented data: expert recovery actions serve as chosen trajectories, while erroneous actions leading to error states serve as rejected trajectories.
We evaluated four training configurations across combinations of SFT and DPO stages, using either optimal-only or MAF-augmented data (detailed in Sec. 4.4.4, Table 2 of the updated manuscript). Our key findings are discussed below.

MAF data is critical for robust recovery: Models warm-started with MAF data consistently outperform those trained on just expert data, even when both variants are subsequently refined with DPO. The combination of using SFT with DPO on the MAF data reaches 92% success rate, demonstrating that preference-based refinement can provide marginal gains when applied to mistake-aware data. Remarkably, extending SFT on MAF data alone achieves 91% success rate, comparable to the full DPO pipeline but with significantly simpler training. This validates that MAF's targeted loss masking effectively teaches recovery behaviours without requiring complex preference optimisation.

These results demonstrate that learning mistake-aware recovery behaviours is the primary driver of robustness, with DPO providing only modest additional improvements. MAF's simplicity and effectiveness make it particularly attractive for practical deployment.

---

### Meta-Review · Area_Chair_iA3L · 2026-01-07

**Summary:**

Synthetic mistake construction may not reflect real failure modes.
The “mistakes” used for training are artificially injected (e.g., random subgoal insertion/removal, suboptimal moves) and may not resemble realistic errors from perception noise, motor drift, memory failures, environmental dynamics, or causal misunderstandings. Reviewers doubt whether recovery learned from these perturbations will transfer to realistic settings.

Mistake-awareness is under-defined:
The paper uses “mistake awareness” as a key contribution but the papers does not provide a systematic definition or categorization of mistake types.  The experiments cover only a narrow class of symbolic planning deviations, and the paper lacks analysis of recovery effectiveness by mistake category (perceptual vs. causal vs. memory vs. motor, etc.), making the concept feel shallow.

The title suggests “learning from mistakes” in the sense of reducing future mistakes, but the method is framed more as error recovery after a mistake occurs. Reviewers argue the title and framing should align with “error recovery/robustness” rather than implying general learning-from-errors.

Impractical assumption:
Assum the mistake step is given in advance.  The method appears to require identifying exactly which step is the mistake ahead of time. Reviewers argue that in realistic deployments it is often unknown where the mistake occurred, so the approach may not be applicable.

Method is incremental.
The approach is characterized as a form of weighted behavior cloning with binary weights (ignore mistaken step vs. keep), which reviewers see as a straightforward heuristic. The reviewers ask for more innovative mechanisms (e.g., learned weighting or value-based reweighting) and stronger justification that masking-based fine-tuning is more than a simple trick.

Unclear relevance to multimodal / embodied integration.
Reviewers ask whether the method is intended to work with sensor-based observations (vision, proprioception) or continuous control domains, and how it would be integrated into larger embodied systems (e.g., robot planners using vision-language inputs). Handling non-symbolic state representations is not addressed.

**Reviewer Concerns:**

The authors evaluate four configurations that vary (i) whether SFT is used, (ii) whether DPO is used, and (iii) whether training data is optimal-only or MAF-augmented (details in Sec. 4.4.4 and Table 2). The results show that MAF-augmented data is the dominant factor for robust recovery: models warm-started with MAF data consistently outperform models trained on expert-only data, even when both are later refined with DPO.

The authors added an evaluation using Direct Preference Optimization (DPO) as a representative preference-based RL method with implicit reward modeling.

Even with the added DPO experiments, several core concerns remain unresolved.

The paper’s evidence remains confined to symbolic/textual proxy environments (Tower of Hanoi, MiniGrid). This does not test the real sources of brittleness in embodied or multimodal agents.

The training mistakes are injected in a narrow way (e.g., subgoal insertion/removal, suboptimal moves), which does not capture realistic error modalities (perceptual, causal, memory, motor).

The method appears to require identifying the erroneous step in advance to apply masking / ignore loss at that step. In real deployments, the agent typically does not know when it made the mistake without an additional detection mechanism. Since mistake detection is not addressed or evaluated, the approach is not operationally complete.

**Reviewer Scores:**

The scores have reflected the quality of this paper.

---

### Decision · Program_Chairs · 2026-01-26

Reject